# Brain Symptoms of Tuberous Sclerosis Complex: Pathogenesis and Treatment

**DOI:** 10.3390/ijms22136677

**Published:** 2021-06-22

**Authors:** Masashi Mizuguchi, Maki Ohsawa, Hirofumi Kashii, Atsushi Sato

**Affiliations:** 1Department of Developmental Medical Sciences, Graduate School of Medicine, The University of Tokyo, 7-3-1 Hongo, Bunkyo-ku, Tokyo 113-0033, Japan; maki111mew@gmail.com; 2Department of Pediatrics, National Rehabilitation Center for Children with Disabilities, Itabashi-ku, Tokyo 173-0037, Japan; 3Department of Neuropediatrics, Tokyo Metropolitan Neurological Hospital, Fuchu, Tokyo 183-0042, Japan; ocean.letter@gmail.com; 4Department of Pediatrics, Graduate School of Medicine, The University of Tokyo, Bunkyo-ku, Tokyo 113-8655, Japan; satoa-ped@h.u-tokyo.ac.jp

**Keywords:** mTORopathy, mTOR inhibitor, TSC, epilepsy, intellectual disability, autism, epileptic encephalopathy

## Abstract

The mammalian target of the rapamycin (mTOR) system plays multiple, important roles in the brain, regulating both morphology, such as cellular size, shape, and position, and function, such as learning, memory, and social interaction. Tuberous sclerosis complex (TSC) is a congenital disorder caused by a defective suppressor of the mTOR system, the TSC1/TSC2 complex. Almost all brain symptoms of TSC are manifestations of an excessive activity of the mTOR system. Many children with TSC are afflicted by intractable epilepsy, intellectual disability, and/or autism. In the brains of infants with TSC, a vicious cycle of epileptic encephalopathy is formed by mTOR hyperactivity, abnormal synaptic structure/function, and excessive epileptic discharges, further worsening epilepsy and intellectual/behavioral disorders. Molecular target therapy with mTOR inhibitors has recently been proved to be efficacious for epilepsy in human TSC patients, and for autism in TSC model mice, indicating the possibility for pharmacological treatment of developmental synaptic disorders.

## 1. Introduction

The mammalian target of the rapamycin (mTOR) system is an essential signal transduction system inherent in all mammalian cells [1] (Figure 1). Its upstream area consists of several branches. One branch receives extracellular signals, such as insulin and insulin-like growth factors (IGFs), then transmits the information via phosphatidylinositol 3-kinase (PI3K) and protein kinase B (AKT). Another branch accepts signals, such as platelet-derived growth factor, nerve growth factor, and epidermal growth factor, then communicates the information via Ras, mitogen-activated protein kinase kinase (MEK), and extracellular signal-related kinase (ERK). The other branches play roles as sensors of the cellular energy status and the availability of amino acids. In the midstream area, these branches merge into a single flow at the TSC1 (hamartin)/TSC2 (tuberin) complex, a negative regulator of the system that inhibits the activities of Ras homolog enriched in brain (Rheb) and mTOR complex 1 (mTORC1) [2,3,4,5,6]. mTORC1 is a target molecule for pharmacological treatment with rapamycin and its derivatives (rapalogs). Downstream of mTORC1, the flow of signal transduction is divided again into multiple branches. One stream promotes protein synthesis via ribosomal protein S6 kinase (S6K) and ribosomal protein S6 (S6) [7], one enhances cap-dependent translation via eukaryotic translation initiation factor-4E (eIF4E)-binding proteins (4EBPs) and eIF4E [8,9,10], and another inhibits autophagy via ULK-51-like kinase 1 (ULK1) [11,12,13]. The mTOR system regulates various cellular functions, such as growth, proliferation, metabolism, and survival/death. In systemic organs, it is critically involved in multiple processes, including neurogenesis [14], nutrition [15], and immunity [16]. In the brain, its roles are essential in cerebral cortical development, synaptic functions, and brain activities, such as learning, cognition, and social functions [17,18,19].

Medical genetic studies have found many congenital disorders caused by a genetic defect in factors of the mTOR system, collectively referred to as mTORopathies. These conditions share the common brain symptoms of cerebral cortical dysgenesis, epilepsy, intellectual disability (ID), and/or autism spectrum disorder (ASD) [19]. Tuberous sclerosis complex (TSC) is a typical mTORopathy inherited in an autosomal dominant fashion. The two causative genes of TSC, *TSC1* (chromosome 9q34) and *TSC2* (16p13.3) [20,21], are located at the crossroad midstream of the mTOR pathway. Most brain symptoms of TSC are manifested by the dysregulation (hyperactivation) of the mTOR system. Since the 1990s, research has rapidly progressed from the identification of the genetic etiology [20,21] and elucidation of the molecular pathogenesis [22,23,24], to the development of molecular target therapies [25,26]. Currently, mTOR inhibitors are widely used in clinical practice to treat patients with TSC. Notably, they are efficacious not only for TSC-associated tumors, but also for some of the TSC brain symptoms, such as epilepsy [27].

## 2. Pathology and Clinical Picture of TSC

### 2.1. Systemic Findings

Recent advances in medical imaging and genetics have widened the clinical spectrum of TSC far beyond the classical triad of facial angiofibroma, epilepsy, and ID [28]. From a pathologic point of view, TSC is characterized by the multifocal occurrence of benign tumors (hamartomas) and focal dysplastic lesions (hamartias) in various organs, such as the skin, brain, eye, heart, lungs, and kidneys [29] (Table 1). In the vast majority of organs, these morphologic lesions, especially tumors, are the sole cause of functional problems, such as dysmorphism, rupture, and pressure to the surrounding normal tissues. In this context, the cerebrum is a remarkable exception as many patients with TSC have brain dysfunction, such as ID and ASD [30,31], without an apparent causal relationship with anatomical lesions. Thus, the clinical findings of TSC can be classified into three groups: (1) hamartoma, (2) focal dysplasia, and (3) brain dysfunction.

With regard to the occurrence/distribution of lesions and the severity of brain dysfunction, these clinical symptoms are remarkably variable among patients, which is also true for familial cases with the same mutation. The genotype–phenotype correlation is reported to be small [32,33,34], and the reason for the inter-case variability remains largely unknown. Each lesion or symptom shows a time course that is clearly age dependent. For instance, cardiac rhabdomyoma arises in the fetal period, facial angiofibroma in childhood, and pulmonary lymphangioleiomyomatosis (LAM) in adulthood (Table 2). Thus, the management of TSC patients requires life-long follow up.

### 2.2. Brain Symptoms

#### 2.2.1. Epilepsy

Many (about 80%) TSC patients have epileptic seizures of variable types. Typical clinical pictures are West syndrome (infantile spasms) in infancy, and focal epilepsy, which may occur in any age group [30,31]. In the vast majority of patients, the epileptic focus is located in or adjacent to a cortical tuber (focal dysplasia of the cerebral cortex). Epilepsy is often resistant to antiepileptic drugs, requiring neurosurgical treatment in many cases.

#### 2.2.2. ID and ASD

The level of intelligence is variable among patients, ranging from normal to profound ID. ID is present in more than half of patients, and ASD in about half. Even patients with normal intelligence have a variety of behavioral, cognitive, and psychosocial problems, which are collectively called TSC-associated neuropsychiatric disorders (TAND) [35].

#### 2.2.3. Brain Tumor

Approximately 10% of patients have subependymal giant cell astrocytoma (SEGA), a benign tumor on the wall of the lateral ventricle. A large SEGA may cause hydrocephalus and clinical signs of increased intracranial pressure. Hydrocephaly is usually ascribed to the occlusion of the foramen of Monro, although the exact mechanism is still poorly understood. For the treatment of SEGA, the first choice is surgical resection of the tumor. Chemotherapy with an mTOR inhibitor, everolimus, is also efficacious, and has recently become another choice of therapy [25].

## 3. Etiology and Pathogenesis of TSC

### 3.1. TSC Gene Mutations and Their Consequences

TSC is caused by various loss-of-function mutations in the two genes, *TSC1* and *TSC2* [33,34]. No hotspot mutation has been reported. A genotype–phenotype relationship has been noted, but it is small. There is no qualitative difference between *TSC1* and *TSC2*. However, *TSC2* mutations tend to show more severe brain symptoms and a larger propensity to develop tumors than *TSC1* mutations [32,33,34].

*TSC1* and *TSC2* encode for tumor suppressors, hamartin and tuberin, respectively. These proteins bind to form a complex and then stabilize each other [2,3]. TSC-associated tumors show a decreased expression in both [36,37]. The TSC1/TSC2 complex is located in the midstream of the mTOR pathway and negatively regulates the activity of the system [2,3,4,5,6] (Figure 1).

In TSC, a decrease in the regulatory function of TSC1/TSC2 causes the chronic hyperactivation of the downstream mTOR system, which affects cellular proliferation [38], migration [39,40], glucose uptake/metabolism [41], and angiogenesis [42], leading to tumorigenesis and dysgenesis.

### 3.2. Germline and Somatic Mutations

In patients with TSC, all the somatic cells have a germline mutation (first hit) in one allele of either the *TSC1* or *TSC2* gene, which can cause haploinsufficiency of the TSC1/TSC2 complex. When an additional somatic mutation (second hit) occurs during mitosis, the function of the TSC1/TSC2 complex becomes null.

Previous studies have demonstrated that TSC-associated tumors (hamartomas) occur according to the two-hit hypothesis [22,23,43]. The second hit is typically a small deletion of either 9q34 (*TSC1*) or 16p13.3 (*TSC2*), causing loss of heterozygosity (LOH), the incidence of which is reportedly high in kidney tumors (renal angiomyolipoma) but low in brain tumors (SEGA).

The genetic mechanisms of TSC-associated focal dysplasia (hamartias) largely remain to be elucidated. In cerebral dysplastic lesions, namely cortical tubers, LOH has not been found [22,23,43,44]. Studies using laser capture microdissection have found point mutations (but not LOH) in abnormal giant cells (astrocyte-like balloon cells and cytomegalic neurons), which are a histopathological hallmark of cortical tubers [24,45]. A cortical tuber comprises a small number of abnormal giant cells with null TSC1/TSC2 function, and a large number of normal-sized neurons/glial cells with haploinsufficiency [46]. The interaction of both cell types may account for the epileptogenicity of cortical tubers (Figure 2).

mTOR inhibitors are useful in the treatment of TSC-associated tumors in the brain, heart, and kidneys (everolimus) [25,26,47], as well as in the skin and lungs (sirolimus) [48,49]. Furthermore, clinical studies have recently shown that mTOR inhibitors are effective not only for tumors, phenotypes predominantly resulting from the second hit, but also for brain dysfunction (epilepsy) caused by a combination of the first and second hits [27]. The molecular basis of the efficacy is that the main stream of the mTOR system is essentially single between the TSC1/TSC2 complex and mTORC1 (Figure 1).

## 4. Brain Dysfunction in TSC

### 4.1. ID, ASD, and Epilepsy

Most TSC patients have a variety of behavioral, cognitive, and/or psychiatric problems, collectively termed as TAND. ID and ASD are the most common, affecting about 80% and 40% of patients, respectively. Notably, the distribution of intelligence quotient (IQ) is bimodal, dividing patients into two groups: profound ID with IQ (less than 30), and normal/subnormal mentality with a slight reduction in average IQ (around 90) [50] (Figure 3). Epidemiologic studies have shown that early-onset epilepsy (represented by West syndrome) is much more common in the former group than in the latter [51,52].

TSC is a common cause of symptomatic ASD, second only to fragile X syndrome in prevalence. The sex ratio (male:female) of entire ASD is 4:1, whereas that of TSC-associated ASD is 1:1 [53,54]. In TSC, ASD is more common in patients with early-onset epilepsy [35,55].

### 4.2. Pathophysiology: Neural and Glial Dysfunction

In TSC-associated epilepsy, the epileptic focus is usually located within or adjacent to a cortical tuber. As described above, a cortical tuber consists of both abnormal giant cells caused by a somatic mutation (second hit) and normal-sized neurons/glial cells with a germline mutation (first hit) only. The abnormal giant cells are abnormal, not only morphologically, but also functionally, as indicated by their immunohistochemical and electron microscopic features [56,57]. Pathologic collaboration (networking) between severely dysfunctional, abnormal giant cells and mildly dysfunctional, normal-sized neurons/glial cells forms the epileptogenic focus. Thus, in the brain, mild dysfunction due to germline mutations (first hit) can cause clinical symptoms, in sharp contrast to other organs where all the symptoms are due to somatic mutations (second hit). There is evidence for the dysfunction of autophagy/cell clearing, which is implicated in epileptogenesis [12,13,58,59].

Previous studies in human patients and animal models, either in vivo or in vitro, have documented various abnormalities of TSC neurons and/or glial cells (Figure 4). For instance, an astrocyte-specific conditional knockout of the *Tsc1* gene impairs the astrocytic transport of glutamate [60]. The expression and function of gamma aminobutyric acid (GABA) receptors are abnormal in abnormal giant cells of cortical tubers [61,62,63,64,65]. Synapse formation is defective in human cortical tubers [66], as well as in cultured neurons of *Tsc1* and *Tsc2* knockout mice [67]. Synaptic pruning is also defective in *Tsc2* knockout mice [68]. There are also abnormalities of synaptic functions, such as impaired long-term potentiation (LTP) and long-term depression (LTD) in the hippocampus of *Tsc1/Tsc2* knockout mice [17,69,70]. These changes, in turn, cause an excitation/inhibition (E/I) imbalance toward hyperexcitation [71]. White matter also shows abnormal findings, which are implicated in the pathogenesis of ASD. Diffusion tensor imaging (magnetic resonance imaging) reveals an abnormal integrity of cerebral white matter, which is improved after pharmacologic treatment with an mTOR inhibitor [72,73]. Taken together, these changes in the morphology and function of neurons and glial cells lead to TSC-associated brain dysfunction: epilepsy, ID, and ASD [71].

### 4.3. Efficacy of mTOR Inhibitors: Animal Experiment

To date, numerous animal models of TSC have been developed and used. One model, Eker rat, occurred spontaneously, whereas all the other models were produced artificially by genetic engineering. For translational research on the treatment of TSC-associated brain functions, we and other investigators often used mice produced by conventional gene knockout, due to their excellent construct validity.

The Eker rat, harboring a germline *Tsc2* mutation, was originally found as a model of the hereditary cancer, renal cell carcinoma [74,75,76]. Compared to human TSC, the brain phenotype of the Eker rat is much milder, with a rare occurrence of cortical tuber-like lesions (focal cortical dysplasia) [77], an absence of epileptic seizures occurring spontaneously, and only a mild deficit in social interaction. Experimental studies introducing a second hit stimulus to the developing brain revealed that irradiation, but not a carcinogen, can increase the incidence of abnormal giant cells [78,79]. When the developing brain was pharmacologically exposed to severe epilepsy, the rat displayed ASD-like social deficit behavior [80].

Except for several *Tsc1*^+/−^ mouse models [81,82], all heterozygous knockout mice for either *Tsc1* or *Tsc2* mutants showed neither cortical tubers nor spontaneous epileptic seizures. Thus, most *Tsc1*^+/−^ and *Tsc2*^+/−^ mice are not suitable as models of TSC-associated epilepsy, or brain symptoms caused by a combination of the first hit (normal-sized neurons and glial cells) and the second hit (abnormal giant cells). However, some of them did have mild, but recognizable, deficits in cognition and behavior [83]. For example, a *Tsc2*^+/−^ mouse showed deficits in some hippocampal functions, such as spatial learning and contextual discrimination. Importantly, these deficits can be reversed by pharmacological treatment with rapamycin [17]. In two other strains, *Tsc1*^+/−^ and *Tsc2*^+/−^, both male and female adult mice showed a deficit in social interaction, an ASD-like phenotype, which again was successfully improved by rapamycin. This improvement in behavior was accompanied by the normalization of gene expression and protein phosphorylation of the mTOR pathway factors [18]. These studies demonstrated that cognitive and behavioral problems in TSC, namely TAND, have not only structural, but also functional aspects, and that the latter can be improved even in adulthood. They also established *Tsc1*^+/−^ and *Tsc2*^+/−^ mice as models of mild TAND in the absence of early-onset epilepsy and illustrated their value in translational research of therapies for TAND.

### 4.4. Efficacy of mTOR Inhibitors: Clinical Trials

Since the beginning of this century, pharmacological treatment with mTOR inhibitors has rapidly progressed, especially for TSC-associated hamartomas: everolimus for brain tumors (SEGA) [25] and renal tumors (AML) [26], and sirolimus (rapamycin) for skin tumors (facial angiofibroma, topical) [48] and pulmonary tumors (LAM) [49]. When treated with mTOR inhibitors, these tumors shrink over several weeks or months. If the treatment continues, the decreased size remains unchanged over years, whereas the tumors often (but not always) grow back again if the treatment is discontinued. A significant difference in the efficacy, or the degree of reduction in size, is noted among patients, or among different types of tumors. For example, brain and kidney tumors decrease in volume, at best, by 70–80%, but never disappear. At worst, the tumor size remains unchanged [25,26], which nevertheless is usually acceptable in clinical practice because the tumors are benign in nature. For some tumors, the reasons for the inter-case or inter-lesion differences have been elucidated. In the treatment of renal AMLs with oral everolimus, the reduction in size is larger in fat-poor tumors than in fat-rich tumors [84]. In the treatment of skin tumors with topical sirolimus, the clinical effect is more rapid and remarkable in facial angiofibromas (those rich in blood vessels and covered by thin skin) than in other types of skin tumors (those poor in blood vessels and/or covered by thick skin) [85].

During clinical trials of oral (systemic) administration mTOR inhibitors, researchers noted their ancillary efficacy on brain dysfunctions, such as a decrease in epileptic seizures and the improvement of ASD symptoms. A phase 3, international clinical trial on the efficacy of everolimus for focal epilepsy (named EXIST-3) was conducted, and successfully demonstrated the usefulness of everolimus in the treatment of TSC-associated epilepsy by showing a clinically meaningful reduction in seizure frequency [27]. On the other hand, the inter-case variability was even larger in epilepsy than in cerebral and renal tumors; epileptic seizures totally disappeared in the best case, whereas they increased by three times in the worst case. With regard to ASD, a Japanese sub-study of EXIST-3 investigated the effects of everolimus on ASD symptoms using the Pervasive Developmental Disorders Autism Society Japan Rating Scale (PARS). Similar to the situation with epilepsy, the effect varied considerably among cases. Of the 11 patients treated by everolimus, 4 showed improvement (a decrease in PARS score by 5 or more) and one worsened (an increase by 5 or more) [86]. With regard to both epilepsy and ASD, the reasons for the large inter-case variability remain unclear. The usefulness of mTOR inhibitors as a therapy for ASD still remains to be established.

## 5. Epileptic Encephalopathy

Drug-resistant epileptic syndromes with onsets in infancy or childhood, such as West syndrome and Lennox–Gastaut syndrome, are often accompanied by the impairment of cognitive/behavioral development, such as ID and ASD, and are collectively termed as epileptic encephalopathies [87]. The etiologic factors of epileptic syndromes are variable, including not only TSC, but also malformation, metabolic error, infection, hypoxia/ischemia, and many others. Electroencephalographically, epileptic encephalopathy is characterized by frequent epileptic discharges that appear not only during seizures, but also persist during the interictal period in the absence of clinical symptoms of seizures. In the brain, there is a continuous, severe epileptic discharge causing subclinical electrical status epilepticus that, in turn, increases the activity of the mTOR pathway [88]. In the developing brains of infants, sustained hyperactivity of the mTOR pathway affects the synthesis of synaptic proteins, leading to an abnormal structure of synapses and excessive synchronicity of the brain network [60,61,62,63,64,65,66,67,68]. These morphologic and functional changes further worsen epilepsy and cause ID, ASD, and other TAND symptoms (Figure 4). Thus, epileptic discharges and mTOR hyperactivity constitute a vicious cycle (Figure 5). The treatment of epileptic encephalopathies includes various antiepileptic drugs, adrenocorticotropic hormone (ACTH)/steroids, gamma-globulin and ketogenic diets [89]. The pharmacologic mechanism of ketogenic diets is partially accounted for by a decrease in glucose and insulin, downregulating the activity of the mTOR pathway [90].

Patients with TSC are particularly at high risk of entering the vicious cycle due to the presence of another gate, a mutation in the *TSC1/TSC2* gene (Figure 5). In the treatment of TSC-associated epilepsy in infancy, the potential efficacy of an mTOR inhibitor is theoretically expected [91], but still remains to be proven [92,93]. On the other hand, the usefulness of vigabatrin, a potent antiepileptic drug that increases the synaptic concentration of GABA, has been established [35]. The pharmacological basis for the efficacy of GABA in TSC is partially accounted for by the defective GABA receptors in cortical tubers described above [61,62,63,64,65]. According to standard protocols for the treatment of epilepsy, vigabatrin is prescribed only when the diagnosis of epilepsy was made on the basis of clinical seizures and electroencephalographic findings. In the case of West syndrome, the start of vigabatrin treatment was typically in mid-infancy. To improve the neurologic outcome of TSC-associated epilepsy of infantile onset, some investigators in Europe have tried early treatment with vigabatrin. In these studies, patients with TSC underwent serial electroencephalography in early infancy and were “prophylactically” prescribed with vigabatrin immediately after the appearance of an epileptic discharge. The results of the preliminary studies were quite promising, with a better outcome at two years of age for both epilepsy and neurocognitive development [94]. These beneficial effects were sustained into childhood [95].

Epilepsy surgery is another way to break out of the vicious cycle of TSC-associated epileptic encephalopathy (Figure 5). Many investigators have demonstrated that early resection of an epileptogenic tuber improves the neurologic outcome, both in terms of the control of epileptic seizures and cognitive/behavioral development, of infants and children with pharmaco-resistant epilepsy [96,97,98].

## 6. Conclusions

Patients with TSC have a mutation in either *TSC1* or *TSC2*, causing a loss of function of the TSC1/TSC2 complex and the dysregulation of the mTOR system. This loss results in haploinsufficiency in a large number of cells with only a germline mutation, but completely nullifies the function in a small number of cells with an additional somatic mutation, such as the tumor cells of SEGA and abnormal giant cells of cortical tubers. Clinically, haploinsufficiency remains harmless, causing neither structural nor functional abnormalities in all systemic organs. However, the brain is the only organ where haploinsufficiency by itself causes a mild, but recognizable, disturbance in cognitive and behavioral function, known as TAND, such as a slight decline of IQ by approximately 10 in human patients (Figure 3), and a decrease in social interaction in *Tsc1*^+/−^ and *Tsc2*^+/−^ mice. If a somatic mutation in the developing brain affects a subset of neurons, glial cells, or their precursor cells, the morphology and/or function of the cells may become severely impaired, as is the case with abnormal giant cells. The pathologic orchestration of the “haploinsufficient cells” and “null cells” may create a synaptic network with a tendency toward a distorted E/I balance. When severe epilepsy actually occurs in infancy, which is the critical period of synaptic formation and pruning, a vicious cycle of epileptic encephalopathy is formed, resulting in severe disability (ID and/or ASD) and further worsening of epilepsy. Recent advances in the studies of TSC have provided clinicians with candidates for robust therapeutic measures to escape from the vicious cycle, such as vigabatrin, mTOR inhibitors, and epilepsy surgery. Further progress in clinical and basic research is needed to significantly improve the neurological outcome and quality of life of infants and children with TSC-associated epilepsy.

## Figures and Tables

**Figure 1 ijms-22-06677-f001:**
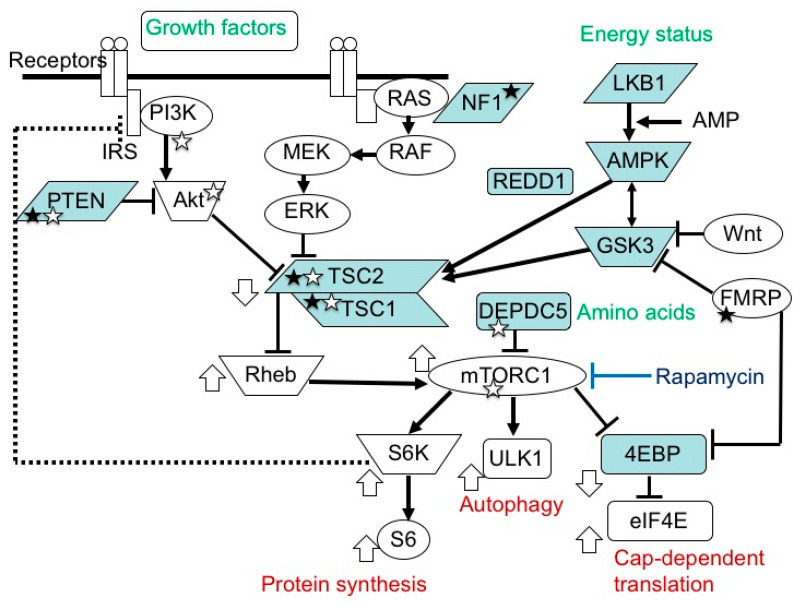
mTOR signaling pathway. In the upstream region, there are two paths, one transmitting signals of growth factors, and another transmitting those of energy status. In the midstream region, they converge at the TSC1/TSC2 complex into one main stream, which divides again at mTORC1 into S6K/S6 (protein synthesis), ULK1 (autophagy), and 4EBP/eIF4E (cap-dependent translation) pathways. The white circles and blue boxes represent factors that activate and inhibit, respectively, the activity of this system. Genetic defects in factors with a white and black star cause epilepsy and ASD, respectively.

**Figure 2 ijms-22-06677-f002:**
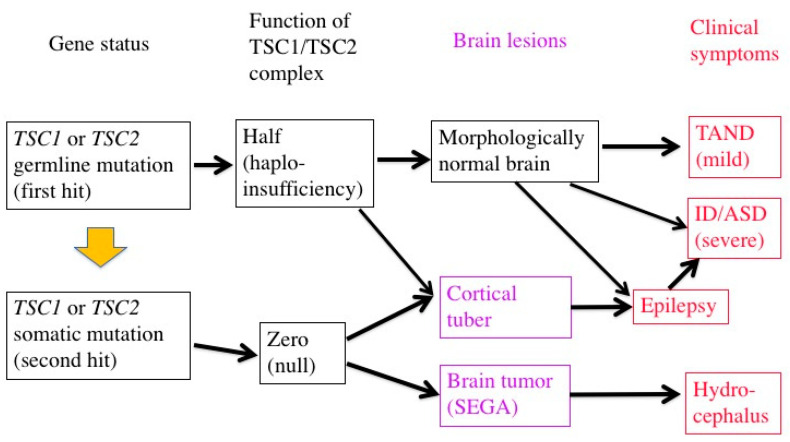
Two-hit hypothesis in a brain with TSC. The first hit, or germline mutation, can cause, by itself, mild TAND symptoms without any morphologic abnormalities. The second hit, or somatic mutation, may produce morphologic lesions (purple), either hamartoma (SEGA) or hamartia (cortical tuber). A tuber consists of a large number of “haploinsufficient” neurons and glial cells of normal size, and a small number of “null” cells of abnormal size and shape, and may become the focus of epileptic discharge. Severe early-onset epilepsy distorts the development of the brain network, leading to severe ID and/or ASD.

**Figure 3 ijms-22-06677-f003:**
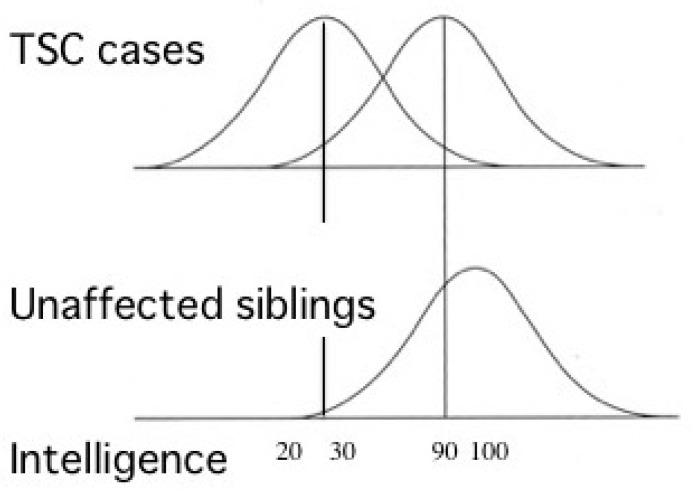
Cognitive function of patients with TSC showing bimodal distribution of IQ (modified from Joinson et al. (2003) [50]). Severe early-onset epilepsy is common in patients with profound ID (IQ < 30), but not in those with normal/subnormal intelligence (IQ around 90).

**Figure 4 ijms-22-06677-f004:**
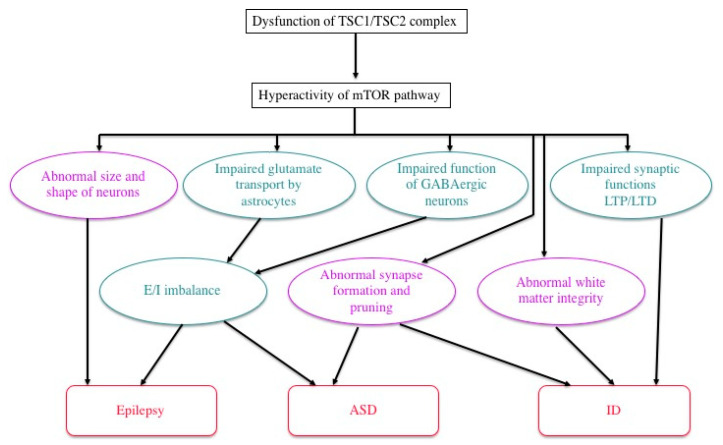
Possible mechanisms of TSC brain symptoms by which a loss of function of the TSC1/TSC2 complex causes a variety of structural (purple) and functional (green) abnormalities, leading to epilepsy, ID, and/or ASD (modified from Napolioni et al. (2009) [71]).

**Figure 5 ijms-22-06677-f005:**
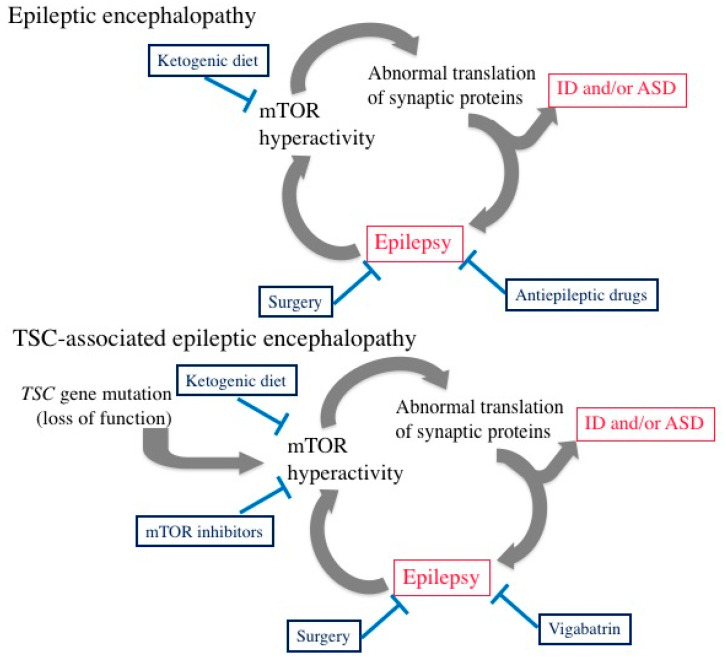
Vicious cycle in the brains of infants with epileptic encephalopathy of any etiology (**top**), and its association with TSC (**bottom**), showing the involvement of the mTOR pathway. Boxes with blue letters indicate therapeutic measures currently available and commonly used.

**Table 1 ijms-22-06677-t001:** Characteristic lesions of TSC (modified from Gomez (1999) [29]).

Organ	Hamartias	Hamartomas
Skin	Hypomelanotic macules Confetti skin lesions	Facial angiofibromas Fibrous cephalic plaque Ungual fibromas Shagreen patch
Brain	Cortical tuber	SEGA Subependymal nodules
Eye	Retinal achromic patch	Retinal hamartoma
Mouth	Gingival fibromas	Dental enamel pits
Lung		Pulmonary LAM MMPH
Heart		Cardiac rhabdomyoma
Arteries	Wall defects/Aneurysm	Renal AML
Kidney	Renal cysts	
Bone	Bone cysts	

**Table 2 ijms-22-06677-t002:** Age-dependent changes of lesions and symptoms in TSC.

Lesion/Symptom	Pathology	Age of Occurrence or Worsening	Department in Charge	Note
Cardiac rhabdomyoma	Tumor	Fetal–neonatal period	Pediatric cardiology	Spontaneous regression in infancy
Cortical tuber	Dysplasia	Fetal–neonatal period/Infancy	Pediatric neurology	Focus of epileptic seizures
Hypopigmented macule	Dysplasia	Fetal–neonatal period/Infancy	Dermatology	
Epilepsy	Brain dysfunction	Infancy/Childhood	Pediatric neurology	Intractable in many cases
ID/ASD	Brain dysfunction	Infancy/Childhood	Pediatric neurology/Psychiatry	
Retinal hamartoma	Tumor	Infancy/Childhood	Ophthalmology	
Renal cyst	Dysplasia	Infancy/Childhood	Pediatrics	
SEGA	Tumor	Childhood/Adolescence	Neurosurgery	Hydrocephalus, potentially fatal
Facial angiofibroma	Tumor	Childhood/Adolescence	Dermatology	
TAND	Brain dysfunction	Childhood/Adolescence	Pediatrics/Psychiatry	
Renal AML	Tumor	Childhood/Adolescence/Adulthood	Pediatrics/Urology	Hemorrhage, potentially fatal
Pulmonary LAM	Tumor	Adolescence/Adulthood	Pulmonology	Predominantly affecting women, potentially fatal

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
