# Peer review of "Brain Symptoms of Tuberous Sclerosis Complex: Pathogenesis and Treatment"

_ijms, 2021, doi:10.3390/ijms22136677_

Round 1

Reviewer 1 Report

In this paper Mizuguchi  et al. reviewed the etiology, pathogenesis and clinical symptoms of TSC, with a special emphasis on mTOR-dependent brain dysfunction. The Authors focused on the vicious cycle between epileptic encephalopathy and mTOR hyperactivity, which in turn may result in abnormal synaptic structure/function and excessive epileptic discharges, thus further worsening epilepsy and intellectual/behavioral disorders in infants with TSC, as well as on the role for mTOR inhibitors as possible therapeutic approaches for treatment these developmental synaptic disorders.

Overall, the paper is well organized and informative.

However, in the opinion of the Reviewer, it would benefit from implementation on some critical aspects regarding  / critical aspects which need a more in depth analysis:

In the Introduction section, as well as in Figure 1, the Authors only mention the role of mTOR complex as an autophagy inhibitor. However, increasing evidence suggests that dysfunction in cell clearing systems, and in particular in the autophagy pathway, due to mTOR hyperactivation, has a key role in the epileptogenic mechanisms.

McMahon et al. Impaired autophagy in neurons after disinhibition of mammalian target of rapamycin and its contribution to epileptogenesis. J Neurosci. 2012 Nov 7; 32(45):15704-14.

Yasin et al.  mTOR-dependent abnormalities in autophagy characterize human malformations of cortical development: evidence from focal cortical dysplasia and tuberous sclerosis.Acta Neuropathol. 2013 Aug; 126(2):207-18.

Di Nardo A, et al.  Neuronal Tsc1/2 complex controls autophagy through AMPK-dependent regulation of ULK1. Hum Mol Genet. 2014 Jul 15;23(14):3865-74.

Limanaqi F, et al. mTOR-Related Cell-Clearing Systems in Epileptic Seizures, an Update. Int J Mol Sci. 2020;21(5):1642.

Wong M Mammalian target of rapamycin (mTOR) inhibition as a potential antiepileptogenic therapy: From tuberous sclerosis to common acquired epilepsies.Epilepsia. 2010 Jan; 51(1):27-36.

Discussing more in depth the implications and translational significance of such alterations in these neurodevelopmental disorders would add significance to the manuscript.

The English language and style are fine. However, there are a few places with wording that needs to be smoothed over. Minor point: there are some grammatical/spelling errors and minor spell check is required.

Reviewer 2 Report

#1. Section2.2.3; Since the mechanism of hydrocephalus by SEGA is still unclear, we cannot simply say the occlusion of the foramen Monro. We have may patients who have SEGAs as if they occlude their foramen of Monro in the MRI without any increased intracranial pressures. The classical mechanism of hydrocephalus, which is too much simple, is obstructive and communication. However, the fact that more complicated causative mechanisms must be involved in hydrocephalus, such as brain compliance and osmotic pressure due to cerebrospinal fluid protein is current stream to understand the mechanism of hydrocephalus. I cannot agree with the sentence about the mechanism of hydrocephalus.
